# Stress and Alcohol Intake among Hispanic Adult Immigrants in the U.S. Midwest

**DOI:** 10.3390/ijerph192316244

**Published:** 2022-12-04

**Authors:** Jacqueline Rodriguez, Lilian Golzarri-Arroyo, Cindy Rodriguez, Gerardo Maupomé

**Affiliations:** 1Department of Global Health, Richard M. Fairbanks School of Public Health, Indiana University–Purdue University Indianapolis, Indianapolis, IN 46202, USA; 2Department of Epidemiology and Biostatistics, School of Public Health Bloomington, Indiana University Bloomington, Bloomington, IN 47405, USA; 3Indiana University Network Science Institute, Bloomington, IN 47408, USA

**Keywords:** Hispanic immigrants, alcohol intake, alcohol use disorder, perceived stress, socioeconomic status, Mexican-Americans, Central-Americans

## Abstract

Alcohol intake and Alcohol Use Disorder (AUD) among recent and very recent Hispanic immigrants are not well characterized, in particular in the context of perceived stress among such groups. The objective of the present study was to shed light on alcohol intake and AUD overall, as well as potential modifications derived from varying levels of stress and socioeconomic status (SES). The study population was immigrants with six or fewer months of having arrived in the American Midwest, and members of their peer networks who had been in the U.S. for 2+ years. We found that AUD and alcohol intake spanned from very high to a considerable proportion who abstained; perceived stress did not have an obvious impact on AUD or alcohol intake. Moreover, neither New vs. Established immigrant statuses, or SES levels, were associated with AUD or alcohol intake. Future research should examine in a more finely-grained approach the components of SES to verify if the complex circumstances of recent immigrants are in fact amenable to SES classification using standard quantification approaches—even using the functional descriptions of the SES surrogates we used.

## 1. Introduction

Latinos/as/x/e or Hispanics are a fast-growing group in the United States (US). The latest data from 2020 showed the Hispanic population had reached 62.1 million, an increase of 23% from the prior decade [1]. Nearly two-thirds of Hispanics living in the US are native-born; however, over one-third are foreign-born immigrants [2]. Social, economic, and cultural factors are associated with health outcomes. Above and beyond socio-economic status (SES), health disparities are shaped by social dynamics including acculturation, comorbid health conditions, social connectivity, and health literacy [3]; yet the relative contributions of these factors differ across populations and groups [4]. In the case of Hispanic immigrants in the U.S., the process of migration and the ensuing unstable social environment may synergize to increase health disparities among first and second+ generation immigrants [5].

Although research regarding alcohol use among Hispanics in the U.S. has expanded over recent years, there are several drawbacks to the current literature. First, many of the studies have primarily focused on Hispanics who have lived in the U.S. for several years [6]. Even among those who share the experience of migrating to the U.S., a distinction should be made between those who are newly arrived and those that are well-established in the country. This is an important consideration as studies have found that length of residence in the U.S. does play a role in health outcomes [7]. Second, since many studies have predominantly examined the influence of acculturation on alcohol use, they might fail to characterize the specific contributing roles and potential impact of other factors [8], such as distinct stressors and SES.

The use of alcohol is prevalent among several populations in the U.S., yet marked differences are noted in levels of alcohol intake and alcohol-related consequences across racial/ethnic groups [9]. According to the National Institute on Alcohol Abuse and Alcoholism, 54.5% of Hispanics reported having at least one drink in the past year compared to 70.3% of Non-Hispanic Whites. Furthermore, 84.2% of Non-Hispanic Whites reported drinking at least one drink in their lifetime, compared to only 67.7% of Hispanics [9]. While this current data shows that Hispanics are less likely to drink alcohol compared to non-Hispanic Whites, data also shows Hispanics who do drink are more likely to consume greater alcohol amounts [9]. Frequent consumption of greater alcohol amounts heightens the risk for the development of Alcohol Use Disorder (AUD), a cluster of conditions often associated with uncontrolled drinking despite awareness of its harmful and negative health effects. AUD broadly encompasses all aspects related to severe alcohol consumption, including abuse, dependence, and addiction [9]. Compared to non-Hispanic Whites, Hispanics in the U.S. have consistently been shown to experience disproportionately greater levels of adverse health problems related to alcohol use, including alcohol-related liver cirrhosis and related mortality [10].

Heightened risk for AUD is influenced by several factors including various categories of stressors [11]. A stressor is defined as any experience or environmental condition that induces stress. Unique to immigrants are the multi-dimensional stressors that may arise in each phase of the immigration process. These comprise challenges faced prior to immigrating, including financial constraints, the anguish of leaving their groups of peers and family, as well as the pressures of adapting to a new culture, language barriers, and potential discrimination they may face once they have arrived [12]. Given these hardships, immigrants are susceptible to the experience collectively referred to as perceived stress [13]. Some studies have proposed that recent immigrants are more likely to have greater levels of cultural stress compared to immigrants who have lived in the U.S. for several years [14], adding to the notion that health outcomes may greatly vary among immigrants based on their time spent in the U.S.

SES is another factor known to play a role in alcohol use patterns. Literature concerning the relationship between SES and alcohol use has generally described an inverse relationship. Individuals with higher income and higher SES drink more than those with lower SES; however, individuals with lower SES often suffer greater alcohol-related consequences [15,16]. Because SES has been found to be closely associated with race/ethnicity, both Hispanics and Blacks often fall in the poorer end of the SES range; those low SES categories are associated with lower levels of schooling and income [4,15].

The limited research concerning alcohol use among Hispanics is further complicated by the sparse literature on recent immigrants. Understanding their drinking behavior and factors influencing drinking patterns are important to frame more accurately interventions and preventive efforts to ameliorate the AUD health burden. Because there is no universal classification for ‘recency’ of immigration, it would be empirically and functionally important to better understand the distribution of AUD if we distinguish between those immigrants who very recently arrived from those who are already further ahead in the process of adapting to a new living environment. The primary objective of the present study was to characterize perceived stress levels and alcohol use patterns among New and Established Hispanic immigrants living in the Midwest U.S. A secondary objective was to assess socioeconomic factors and their influence on perceived stress levels.

## 2. Materials and Methods

### 2.1. Procedures

All subjects gave their informed consent for inclusion before they participated in the study. The study was conducted in accordance with the Declaration of Helsinki, and the protocol was approved by the Indiana University Ethics Committee (#1703740862).

Data were collected from 2017 to 2021 in a larger longitudinal study to assess the health perceptions and experiences of Hispanic immigrants living in the state of Indiana, through an initial baseline questionnaire and additional post-enrollment follow-ups performed at 6, 12, and 18 months [17]. Participants were recruited by several engagement methods largely described beforehand in the literature: recruitment sites were primarily Hispanic-serving community organizations and faith-based organizations although businesses catering to the Hispanic community and relevant Facebook, TV and radio media figured as well. A total of 367 base individuals were recruited as ‘recent’ or ‘New’ immigrants who had been living in the U.S. for 6 months or less. An additional 180, designated ‘Established immigrants’, were recruited from the social networks of the recent immigrants: they had to have lived in the U.S. for a minimum of two years. Admittedly, such distinction is an arbitrary separation between recently arrived (within six months of participating in the baseline wave) and Established immigrants, who had been residing in the U.S. for at least two years by the start of the baseline wave. There is no universal categorization to quantify the recency of immigration.

The present analysis utilized only baseline data to explore the relationship between alcohol use and perceived stress. Data collection for the baseline wave began in August 2018. Baseline data were collected prior to the full-blown onset of the SARS-CoV-2 pandemic and concluded exactly when the social distance public health mandates were enacted in March 2020. Therefore, perceptions about AUD, acculturation, stressors, and SES collected for the present research were unlikely to have been modified due to the social shifts ascribable to the pandemic. However, data collection took place during the Trump federal administration (2017–2021), arguably a time of hostility toward immigration—in particular for immigrants from Latin America.

We explicitly excluded questions about documented status (documented vs. undocumented) to ensure we would not incur a selection bias making it more likely study participants had a documented status. The study design, our advertising, and the type of IRB approval pointed out that we would not seek information about the documented status.

Survey and assessment materials were translated and available in both English and Spanish, with participants having the option to complete the questionnaires in their preferred language.

### 2.2. Measures

Alcohol use. Alcohol use and severity were assessed using the Alcohol Use Disorders Identification Test (AUDIT). The AUDIT screens for alcohol frequency, potential alcohol dependence, and alcohol-related problems in a series of 10 self-reported items [18]. Each response is scored on a Likert scale from 0 to 4. Responses are then summed with a possible range of scores from 0 to 40, with higher scores indicating greater alcohol use.

Perceived stress. Stress levels were measured using the 10-item Perceived Stress Scale (PSS-10). The PSS-10 measures feelings and general perceptions of stress. More specifically, items are aimed to assess the extent to which respondents consider their lives to be unpredictable, uncontrollable, and overloading within the past month [13]. Responses are summed across all items, with final scores ranging from 0–40. These scores are categorized as low, moderate, or high perceived stress.

### 2.3. Data Analysis

Descriptive statistics were calculated for demographic variables separately by New immigrant or Established immigrant categories. To explore the relationship between AUDIT and perceived stress, a linear regression was performed. We also included their immigrant status (New or Established) as a moderator to test for a different relationship between AUDIT and perceived stress. Demographics were included in the models as covariates (gender, age, and marital status). Assumptions were tested and log transformations were performed as needed. Analyses where AUDIT is included only applied to participants that responded that they drank at least “Monthly or less frequently”.

To explore the influence of socioeconomic status on stress we ran ANOVA tests. As we did before, we included immigration status as a moderator to see if it affected the relationship between socioeconomic status and perceived stress. Analyses were conducted using R version 4.1.2 (R Core Team, 2021, Vienna, Austria).

## 3. Results

The present analysis included 547 Hispanic immigrants overall (367 New and 180 Established). Participants were male (*n* = 177, 32.4%) and female (*n* = 370, 67.6%) ranging in age from 18–82, with a mean age of 34.4 (SD = 11.2). A large percentage of participants (n = 315, 57.6%) were from Mexico whereas the remaining immigrants (n = 232, 42.4%) were Central American, from El Salvador, Guatemala, or Honduras. Table 1 displays the demographics of the study population.

For participants who drank at least “Monthly or less frequently”, they had a mean AUDIT of 2.39 (SD = 3.58). There were no significant differences between immigrant statuses (*p* = 0.521). New immigrants had a mean AUDIT score of 2.27 (SD = 3.49), and for Established immigrants, AUDIT was 2.60 (SD = 3.74).

Overall mean Perceived Stress for the participants was 13.6 (SD = 7.15); there was a significant difference (*p* < 0.001) between New Immigrants and Established immigrants, where the first ones had a higher level of stress at 14.3 (SD = 7.31) compared to established ones 12.2 (SD = 6.58).

For participants who drank, there was no statistically significant relationship between AUDIT and Perceived Stress (95%CI (−0.02, 0.01), *p* = 0.484). When testing the moderation of immigrant status, the interaction between immigration status and perceived stress was not statistically significant (unadjusted, 95%CI (−0.05, 0.02), *p* = 0.286). When adjusted for age, gender and marital status, the interaction was not statistically significant (adjusted, 95%CI (−0.05, 0.02), *p* = 0.293). In Figure 1 (plotting AUDIT vs. Perceived Stress) we found a lack of relationship by overall immigration status or by New or Established: there were participants with high values for AUDIT regardless of their score of perceived stress.

As a secondary research question, we explored the relationship between SES with perceived stress using four different variables to describe SES (Table 2). When all participants (New and Established) were asked if in the past six months they have had money to meet their needs, there were statistical differences between those who responded, ‘Just Enough’ (*p* < 0.001) and ‘More than enough’ (*p* = 0.016) versus those that said ‘Not Enough’; the latter had higher perceived stress (Table 2). However, when we included immigration status as moderator, only ‘Not Enough’ vs. ‘Just Enough’ was significant (*p* = 0.004) for New immigrants where the ‘Not Enough’ group had higher stress. For Established immigrants, there were no significant differences between groups (Figure 2a).

For the question ‘In the past six months, how difficult it was to pay bills’, for the complete sample across immigration statuses, ‘Somewhat Difficult’, ‘Not Very Difficult’ and ‘Not at all Difficult’ were statistically different on stress scores, compared with ‘Very difficult’ (*p* = 0.025, *p* < 0.001, *p* < 0.001, respectively) (Table 2). ‘Very difficult’ had higher perceived stress scores than the other groups (Figure 2b). When including immigration status as moderator, ‘Very difficult’ vs. ‘Not very difficult’, ‘Very difficult’ vs. ‘Not at all difficult’, ‘Somewhat Difficult’ vs. ‘Not at all Difficult’ were still statistically significant for both New and Established immigrants. ‘Very Difficult’ vs. ‘Somewhat Difficult’ was only statistically different on perceived stress for Established immigrants (*p* = 0.036), not for New immigrants. The remainder of the comparisons (‘Somewhat Difficult’ vs. ‘Not Very Difficult’ and ‘Not Very Difficult’ vs. ‘Not at all Difficult’) were not significant for any of the immigrant groups (Figure 2b).

When asked if in the past six months they have had enough money for food, there was a statistical difference in perceived stress for the complete sample across immigration statuses between ‘Often’ vs. ‘Never’ (*p* < 0.001) (Table 2). When looking at immigration status differences, for both New and Established immigrant groups there were significant differences in stress derived from the ability to purchase food between ‘Often’ vs. ‘Never’, ‘Sometimes’ vs. ‘Never’, and ‘Rarely’ vs. ‘Never’ (Figure 2c). ‘Often’ had the highest perceived score and ‘Never’ the lowest.

Lastly, when asking ‘In the past six months, did you cut meals because you didn’t have enough money’, for the complete sample there was only a significant difference in perceived stress between ‘Often’ and ‘Never’ (*p* < 0.001) (Table 2), with ‘Often’ having higher perceived stress than Never (Figure 2d). For New and Established immigrants there were differences in their stress between ‘Often’ vs. ‘Never’, and ‘Sometimes’ vs. ‘Never’. Only for New immigrants, there was significant differences in perceived stress between ‘Rarely’ and ‘Never’ (Figure 2d).

## 4. Discussion

The present study examined whether levels of perceived stress were linked to AUD and whether differences existed between newly arrived Hispanic immigrants and Hispanic immigrants who had been established in the U.S. for a minimum of two years. The results from our sample did not find a statistical significance between AUD and perceived stress. Regarding the length of time in the U.S., no significant differences were found between New versus Established. These findings suggest that while a portion of the sample did consume alcohol at levels considered to be potentially hazardous and at risk for AUD, perceived stress was not a factor modifying alcohol use. The fraction of participants at potential risk for AUD was considerably low as more than half of the participants in the study abstained from alcohol use entirely. This is consistent with the literature, with prior studies documenting that immigrants overall were less likely to consume alcohol compared to their U.S.-born counterparts and considerably less likely to develop AUD [19,20]. This may further be explained in the literature by the “Healthy Immigrant Effect”, also referred to as the “immigrant paradox”. In short, the paradox depicts immigrants having an advantage in health outcomes compared to their native-born counterparts, despite facing greater disadvantages [19,20]. Researchers have described the paradox as a protective relationship, in which recent immigration serves as a protective factor that decreases with increased length of stay and in subsequent generations [19].

The influence of SES on perceived stress levels was found to be inconclusive. Using variables indicating having enough money to meet basic needs did suggest various levels of stress, even when New or Established immigration status was included as a moderator. One possible explanation for this could have been how comprehensive was our measurement of SES for these two sets of participants: SES encompasses several factors including education level, occupation, and social support [21]. Since the present study focused primarily on financial factors, it may be a limited representation of SES. It is generally accepted that the multidimensional and complex nature of SES makes it difficult to accurately measure it [22]. Prior research has documented that Hispanics, along with African Americans, experience the lowest levels of financial literacy compared to other ethnic groups [23]. As such, Hispanic immigrants may lack a comprehensive perspective of their income, thereby potentially misrepresenting SES [24]. Further exploring the link between SES and alcohol use, more than half of participants in the study reported their household income within the past year as less than $50,000, well below the US national average; this trend resembles prior findings from research carried out in Hispanics in Indiana [25,26,27,28,29]. Considering such reports of low income, the decreased frequency of alcohol consumption is consistent with other findings that have reported that individuals with lower SES consume far less alcohol than those with higher SES—but are more adversely affected by its use [15]. Future research is needed to examine the interaction of other SES components and their relationship to alcohol use and drinking patterns.

A few limitations of the present study should be noted. First, given its cross-sectional design, causal relationships may not be inferred, only associations. Second, data for the study were collected through self-reported questionnaires and therefore vulnerable to recall bias and error by participants. The uncertainty surrounding self-reported alcohol intake is a constant challenge in research, since some participants may purposefully under-represent their intake [30,31]. Moreover, the networks of newer immigrants likely had family members: such familial association for drinking habits/AUD could partially explain why there was no difference between the groups. Lastly, the findings cannot be generalized to all Hispanics living in other regions of the U.S. However, a distinct strength of the present study is that it documents health features in the Midwest U.S., a region that became an emerging immigration gateway from the 1990s to date [32]. Most immigrants from Latin American countries commonly arrived from Mexico, now complemented by a fast-growing Central American population [33].

## 5. Conclusions

Despite its limitations, the present study contributes to the limited research concerning alcohol use among Hispanic immigrants living in the U.S. Results indicated that perceived stress levels were not associated with greater alcohol use severity, a contrast to prior studies [34] that merits further exploration. New immigrants were largely just as likely to drink alcohol and fall in an AUD category as Established immigrants. Moreover, the findings from the present study bring attention to the complexity of the SES construct in the context of alcohol use. Given the increasing migration of Hispanic immigrants to the U.S.—particularly in a new gateway region such as the Midwest—future research is needed to further characterize the alcohol use patterns among New arrivals and Established immigrants who are settled in, to better understand and serve the healthcare needs of this population.

## Figures and Tables

**Figure 1 ijerph-19-16244-f001:**
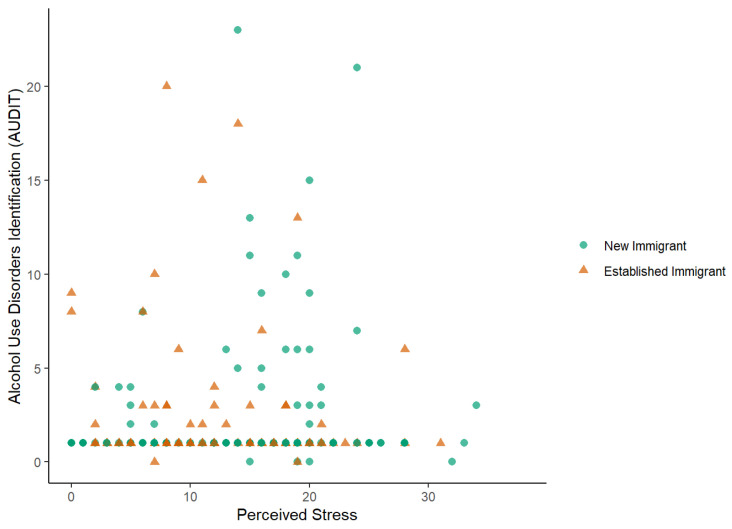
Correlation Plot across AUDIT Scores vs. Perceived Stress Scores, for New and Established Immigrants.

**Figure 2 ijerph-19-16244-f002:**
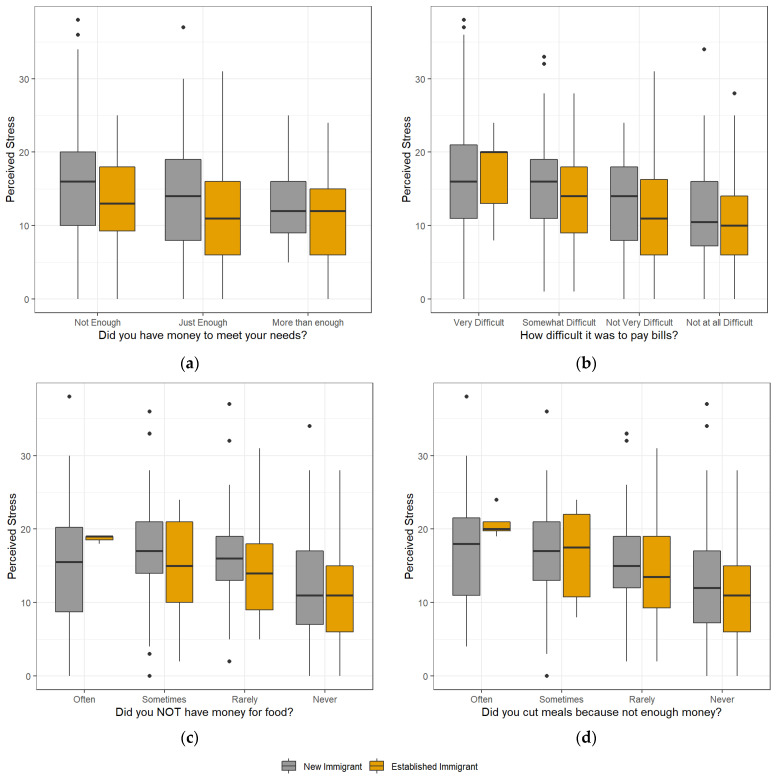
Boxplots for Perceived Stress in the Past Six Months across Socioeconomic Variables by Immigration Recency. Black circles are the default outliers when creating a boxplot [1.5 times the interquartile range above third quartile and below the first quartile]. (**a**) Did you have money to meet your needs? (**b**) How difficult it was to pay bills? (**c**) Did you not have money for food; (**d**) Did you cut meals because not enough money?

**Table 1 ijerph-19-16244-t001:** Participant Demographics.

	New Immigrant (*n* = 367)	Established Immigrant (*n* = 180)	*p*-Value
Nationality, *n* (%)			
Central American	175 (47.7)	57 (31.7)	<0.001
Mexican	192 (52.3)	123 (68.3)	
Gender, *n* (%)			
Female	250 (68.1)	120 (66.7)	0.807
Male	117 (31.9)	60 (33.3)	
Age (years)			
Mean (SD)	33.9 (11.5)	35.4 (10.4)	0.146
Median [Min, Max]	32.0 [18.0, 82.0]	35.0 [18.0, 64]	
Marital Status, *n* (%)			
Single	135 (36.8)	59 (32.8)	0.092
Married/living as	204 (55.6)	108 (60.0)	
Divorced/separated	18 (4.90)	11 (6.11)	
Widowed	9 (2.45)	0 (0)	
Domestic partnership	1 (0.272)	2 (1.11)	
Planned length of stay, *n* (%)			
Permanently	96 (26.8)	85 (47.2)	<0.001
Less than 1 year	43 (12.0)	10 (5.56)	
1–3+ years	11 (3.08)	2 (1.11)	
Unknown	208 (58.1)	83 (46.1)	
Missing data	9 (2.5)	0 (0)	

**Table 2 ijerph-19-16244-t002:** Results from Linear Regression Models on Perceived Stress.

Characteristic	Beta	95% CI ^1^	*p*-Value
Previous six months: money to meet needs			
Not enough	-	-	
Just enough	−2.4	−3.6, −1.1	<0.001
More than enough	−3.1	−5.7, −0.60	0.016
Previous six months: difficulty to pay bills			
Very difficult	-	-	
Somewhat difficult	−1.7	−3.2, −0.22	0.025
Not very difficult	−3.8	−5.6, −2.0	<0.001
Not at all difficult	−5.1	−6.7, −3.6	<0.001
Previous six months: did not have money for food			
Often	-	-	
Sometimes	0.53	−1.7, 2.8	0.643
Rarely	−0.04	−2.5, 2.3	0.972
Never	−4.5	−6.6, −2.4	<0.001
Previous six months: cut meals because of not enough money			
Often	-	-	
Sometimes	−1.3	−4.1, 1.6	0.385
Rarely	−2.9	−5.9, 0.15	0.063
Never	−5.8	−8.5, −3.1	<0.001

CI ^1^ = Confidence Interval.

## Data Availability

The data results that were analyzed for the study are not yet publicly available at the time this manuscript was prepared. Data will be available upon request from the corresponding author.

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
