# Peer review of "Stress and Alcohol Intake among Hispanic Adult Immigrants in the U.S. Midwest"

_ijerph, 2022, doi:10.3390/ijerph192316244_

Round 1
Reviewer 1 Report
Thank you for this interesting paper examining the cross-sectional relationship between alcohol use (measured by the AUDIT questionnaire) and stress (measured by the Perceived Stress Scale), with a secondary examination of new vs established immigration status as a moderator.
1) I would recommend removing the term “off the boat” as it could be considered derogatory.
2) I wonder if recruitment of “established immigrants” from the networks of newer immigrants means that people in the two groups were family members. If so, could familial association with drinking habits/AUD explain why there was no difference in alcohol use patterns between the groups?
3) Perhaps before describing the relationship between AUDIT score and PSS, it would be good to describe the distribution of AUDIT and PSS scores, either in the text and/or in a figure or table.
4) You start off the discussion with a sentence about the relationship between PSS and AUDIT, but most of your results section is devoted to the relationship between PSS and SES questions. Perhaps consider clarifying your objectives. Consider what your primary objective is and what secondary objectives are. Then you can organize the results section accordingly. Given the title of the article, I assumed the focus of the discussion would be the relationship between stress and alcohol use among immigrants, examining new/established status as a moderator.
5) Discussion line 226—I don’t think it’s appropriate to assume that the null relationship in your analysis is due to immigrant ignorance
6) Part of the reason for no relationship between increasing stress and AUDIT score might be the healthy migrant effect which might be protective for recent immigrants (https://www.ncbi.nlm.nih.gov/pmc/articles/PMC8606270/#:~:text=The%20%E2%80%9CHealthy%20Immigrant%20Effect%E2%80%9D%20(,to%20mental%20health%20(MH).) Consider adding a reflection on this in your discussion.
Author Response
November 28, 2022
Dear Reviewer,
The co-authors and I are most grateful for the valuable comments provided. Below please find a detailed account of the changes made to the manuscript (highlighted in yellow) to render it ready for a new editorial evaluation.
Reviewer 1
Thank you for this interesting paper examining the cross-sectional relationship between alcohol use (measured by the AUDIT questionnaire) and stress (measured by the Perceived Stress Scale), with a secondary examination of new vs established immigration status as a moderator.
- I would recommend removing the term “off the boat” as it could be considered derogatory.
Response: In the new version of the manuscript we eliminated this entry.
- I wonder if recruitment of “established immigrants” from the networks of newer immigrants means that people in the two groups were family members. If so, could familial association with drinking habits/AUD explain why there was no difference in alcohol use patterns between the groups?
Response: Most likely there were some relatives. We added this observation as a limitation of the study.
- Perhaps before describing the relationship between AUDIT score and PSS, it would be good to describe the distribution of AUDIT and PSS scores, either in the text and/or in a figure or table.
Response: We described the distribution of both AUDIT and PSS scores following Table 1:
For participants who drank at least “Monthly or less frequently”, they had a mean AUDIT of 2.39 (SD=3.58). There were no significant differences between immigrant status (p=0.521). New immigrants had a mean AUDIT score of 2.27 (SD=3.49), and for Established immigrants, AUDIT was 2.60 (SD=3.74).
Overall mean of Perceived Stress for the participants was 13.6 (SD=7.15); there was a significant difference (p<0.001) between New Immigrants and Established immigrants, where the first ones had a higher level of stress 14.3 (SD=7.31) compared to established ones 12.2 (SD=6.58).
- You start off the discussion with a sentence about the relationship between PSS and AUDIT, but most of your results section is devoted to the relationship between PSS and SES questions. Perhaps consider clarifying your objectives. Consider what your primary objective is and what secondary objectives are. Then you can organize the results section accordingly. Given the title of the article, I assumed the focus of the discussion would be the relationship between stress and alcohol use among immigrants, examining new/established status as a moderator.
Response: We clarified primary and secondary objectives in the Introduction and also reiterated this in the Results. A majority of the results addresses PSS and SES as we were seeking to further illuminate the relationship between alcohol (AUDIT) and perceived stress (PSS) – or lack thereof. We chose the next step to be looking at what were sources of stress and what differences existed between immigration status.
The primary objective of the present study was to characterize perceived stress levels and alcohol use patterns among New and Established Hispanic immigrants living in the Midwest US. A secondary objective was to assess socioeconomic factors and its influence on perceived stress levels.
- Discussion line 226—I don’t think it’s appropriate to assume that the null relationship in your analysis is due to immigrant ignorance
Response: -- now lines 249 and 250. We respectfully point out that a considerable fraction of immigrants in these circumstances do not know how much money they made in the previous year. Some do not file tax returns or work in odd jobs – even besides some salary in a regular job. Other honestly do not know the aggregate: they know the past month made rent, and that this week they had resources to obtain school supplies. We have simplified the observation to make it more neutral:
As such, Hispanic immigrants may lack a comprehensive perspective of their income, thereby potentially misrepresenting SES [24].
6) Part of the reason for no relationship between increasing stress and AUDIT score might be the healthy migrant effect which might be protective for recent immigrants (https://www.ncbi.nlm.nih.gov/pmc/articles/PMC8606270/#:~:text=The%20%E2%80%9CHealthy%20Immigrant%20Effect%E2%80%9D%20(,to%20mental%20health%20(MH).) Consider adding a reflection on this in your discussion.
Response: In the new version of the manuscript, we included the healthy migrant effect (immigrant paradox) as described in lines 233-238:
This may further be explained in the literature by the “Healthy Immigrant Effect”, also referred to as the “immigrant paradox”. In short, the paradox depicts immigrants having an advantage in health outcomes compared to their native-born counterparts, despite facing greater disadvantages [19,20]. Researchers have described the paradox as a protective relationship, in which recent immigration serves as a protective factor that decreases with increased length of stay and in subsequent generations [19].
Sincerely,
Dr. Maupome, for the authors
Reviewer 2 Report
This paper was written very well with no major flaws. The topic area is very interesting. The results are surprising but you have clearly acknowledged the limitations of the study and the areas for future research.
The background is very informative but could benefit with some more details to the readers who are not familiar with alcohol consumption levels in the United States. Details on consumption levels of Hispanics and non-Hispanic whites from the literature should be mentioned to give the reader better understanding of the differences between these two groups.
Author Response
November 28, 2022
Dear Reviewer,
The co-authors and I are most grateful for the valuable comments provided. Below please find a detailed account of the changes made to the manuscript (highlighted in yellow) to render it ready for a new editorial evaluation.
Reviewer 2
This paper was written very well with no major flaws. The topic area is very interesting. The results are surprising but you have clearly acknowledged the limitations of the study and the areas for future research.
The background is very informative but could benefit with some more details to the readers who are not familiar with alcohol consumption levels in the United States. Details on consumption levels of Hispanics and non-Hispanic whites from the literature should be mentioned to give the reader better understanding of the differences between these two groups.
Response: We added a brief statement in the Introduction to provide general context of alcohol use in the US. We also added specific data on consumption levels of Hispanics compared to non-Hispanic whites to further illustrate differences as stated in Line 49:
Use of alcohol is prevalent among several populations in the US, yet marked differences are noted in levels of alcohol intake and alcohol-related consequences across racial/ethnic groups [9]. According to the National Institute on Alcohol Abuse and Alcoholism, 54.5% of Hispanics reported having at least one drink in the past year compared to 70.3% of Non-Hispanic Whites. Furthermore, 84.2% of Non-Hispanic Whites reported drinking at least one drink in their lifetime, compared to only 67.7% of Hispanics [9].
Sincerely,
Dr. Maupome, for the authors
Reviewer 3 Report
Abstarch
- it's ok and presents all results of the study
Introduction
- needs some minor corrections
Line 39 - should include what patterns of alcohol they refer to
Line 57 - what health issues? Should be reported
Line 75 - why only now talk about black people?…no reason if only Hispanics were talked about until now
Results
- I m just not clear if legal and non-legal immigrants would have the same results and if this was accounted for
The discussion needs more comparations with other studies, its a bit poor
Author Response
November 27, 2022.
Dear Reviewer,
The co-authors and I are most grateful for the valuable comments provided. Below please find a detailed account of the changes made to the manuscript (highlighted in yellow) to render it ready for a new editorial evaluation.
Reviewer 3
Introduction
- needs some minor corrections
Line 39 - should include what patterns of alcohol they refer to
Response: Patterns refer to alcohol consumption levels. We omitted the word pattern to minimize confusion.
Line 57 - what health issues? Should be reported
Response: The specific health issues have now been included now in Line 63
including alcohol-related liver cirrhosis and related mortality.
Line 75 - why only now talk about black people?…no reason if only Hispanics were talked about until now
Response: We respectfully point out that the citations we used reached conclusions for both minoritized groups. We quote both to provide context and for completeness.
Results
- Im just not clear if legal and non-legal immigrants would have the same results and if this was accounted for
Response: This may be a significant feature but for ethical and scientific reasons we decided to not collect this datum. We explicitly excluded questions about immigration status (documented vs undocumented). This feature was part of our PAs and ads, to ensure we would not promote a selection bias making it more likely study participants had a documented status. In this way, the study design, our advertising, and the type of IRB approval we obtained pointed out that we excluded questions about immigration status (documented vs undocumented).
We explicitly excluded questions about documented status (documented vs. undocumented) to ensure we would not incur a selection bias making it more likely study participants had a documented status. The study design, our advertising, and the type of IRB approval pointed out that we would not seek information about documented status.
The discussion needs more comparations with other studies, it’s a bit poor
Response: We respectfully point out the various sources in the Discussion to compare other studies with our findings. The studies we cited are what is available in the current literature, which is admittedly rather sparse.
Sincerely,
Dr. Maupome, for the authors
Round 2
Reviewer 1 Report
The updated version is much improved. Thank you for your responsiveness to the reviewer comments.